# A New Approach for Determining Rubber Enveloping on Pavement and Its Implications for Friction Estimation

Di Yun [1,2], Cheng Tang [3], Ulf Sandberg [4], Maoping Ran [1], Xinglin Zhou [1], Jie Gao [5] and Liqun Hu [2,*]

[1] School of Automobile and Traffic Engineering, Wuhan University of Science and Technology, Wuhan 430081, China; yundi@wust.edu.cn (D.Y.)

[2] Key Laboratory of Pavement Structure and Material of Transportation Industry, Chang'an University, Xi'an 710064, China

[3] School of Civil Engineering and Transportation, South China University of Technology, Guangzhou 510641, China; ct_tangcheng@mail.scut.edu.cn

[4] Swedish National Road and Transport Research Institute (VTI), SE-581 95 Linköping, Sweden; ulf.sandberg@vti.se

[5] School of Civil Engineering and Architecture, East China Jiao Tong University, Nanchang 330013, China

[*] Correspondence: hlq123@163.com

**Abstract:** The depth to which the pavement texture is enveloped by the tire tread rubber (d) is an important parameter related to contact performance. This study presents a new method (S-BAC), which relies on the ratio between the real contact area and the nominal tire-pavement contact area (S) and the bearing area curve (BAC), to measure the depth on pavements. The tire-pavement contact was simulated by contact between a non-patterned rubber block and pavement specimens. After analyzing the affecting factors, the new method was compared with previous methods by the d values and the application on the relationship between pavement texture parameters and friction. The results reveal that though there is a linear regression between the d obtained with the S-BAC and previous methods, the d values obtained with different methods differ. Applying the S-BAC method can strengthen the relationship between texture parameters and friction more than other methods.

**Keywords:** pavement texture; enveloping; rubber penetration depth; contact area; bearing area curve; friction coefficient

## 1. Introduction

The pavements have a multiscale surface roughness, it is practically impossible for the tires to establish complete contact with all parts of the pavement surface. The partial contact of the tire tread rubber with the pavement makes it difficult to find a meaningful relationship between performances on the pavement surface and simple texture indicators [1,2]. Therefore, for a better description of the pavement surface influence on the tire/pavement interaction, it is essential to find methods by which to quantify the enveloping effect of the tire rubber on the pavement surface.

### 1.1. Enveloping Profile and Rubber Penetration Depth

Assuming that when the tire load (*F*) is reduced, the rubber block only touches the pavement at the top of the highest asperity, which here is denoted as a height hmax (Figure 1). Increasing the force *F*, the average plane of the bottom surface of the rubber block will move downwards by a distance (*d*), and the rubber will envelop part of the pavement texture. In this case, the deformation on the bottom surface of the rubber block refers to the enveloping profile (red line). The distance between the highest asperity and the lowest point enveloped by the rubber is defined as the rubber penetration depth (*d*). Both the enveloping profile and rubber penetration depth can help explain how the tire deforms into the space between aggregate particles, how the aggregate penetrates tread rubber, and can indicate the partial contact characteristics between the tire and the pavement.

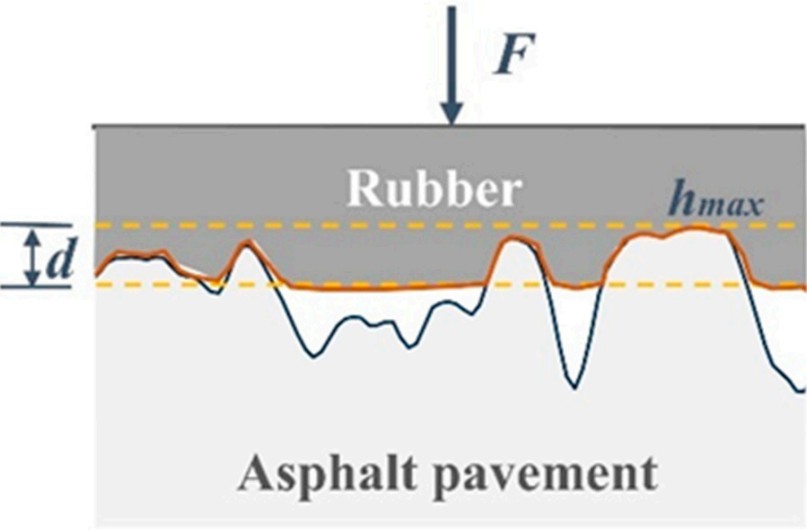

**Figure 1.** The enveloping profile (red line) and the rubber penetration depth.

*1.2. Status of Current Research on Rubber Enveloping of Pavement Texture*

The methods to obtain the penetration depth and to model the enveloping profile can be classified into three categories: physical methods that depend on the laws on which the objects move and behave through space and time, test methods that always rely on experiments, and empiric-mathematical methods, which obtain the enveloping profile by mathematical calculation, while the critical parameters in the calculation process are obtained from testing and statistics.

(1) Physical methods: Clapp and Fong, Klein, and Hamet et al. proposed the solution for contact depth based on elastic half-space deformation under linear load [3]; Persson submitted a multi-scale contact mechanism that excludes arbitrary surface roughness without any prior [4]. Dubois et al. divided the pavement texture into several scale ranges and considered the related parts in terms of contact [5]. Kane and Edmondson described the viscoelastic constitutive relationship of tread rubber using the Kelvin model. They studied the rubber enveloping on pavement texture based on the equal of the integral of interfacial stress and vertical load [6].

However, the application of theoretical models in the tire-road contact must be verified [7]. Mahboob Kanafi and Tuononen [8] and Xiao et al. [9] calculated the average penetration depth based on Persson's contact mechanism with conditional assumptions. The numerical simulation has also been used to study tire-road contact. Srirangam [10] and Zheng et al. [11] integrated the pavement texture into tire-road contact FEA simulation. However, the meshing placed high requirements on the computing ability of computers due to the pavement roughness.

(2) Test method: Matilainen and Tuononen [12] embedded sensors between the rim and the tread, but the buffering effect from the tread rubber weakened the perception of the embedded sensor on the enveloping. Wang et al. [13] tested the height changes of the rubber block surface on the specimens and calculated the penetration depth by subtracting the rubber compression from the height changes. The solution of the rubber compression depended on the elastic assumption. Du et al. [14], Hartikainen et al. [15], and Woodward et al. [16] respectively observed the peeling off of colored powder, asphalt, or paint on the aggregate surface under the rubber to obtain the enveloped area. However, the wear process required more loads, so these methods underestimated the envelope area. Ejsmont and Sommer [17] evaluated the depth of tire tread deformation by casting an imprint of tire-road contact in self-vulcanizing rubber. Chen [18], Lu et al. [19], and Yang et al. [20] tested the skid resistance and the corresponding pavement texture, took the depth corresponding to the good correlation between the texture indicators and the skid resistance as the depth of the enveloped pavement profile. Chen et al. [21], Wang et al. [22], Gao et al. [23],

and Yu et al. [24] used Fuji Prescale film to test tire-pavement contact characteristics. However, the Prescale film directly obtained is point cloud data, from which extracting indicators reflecting contact characteristics is still a problem.

(3) Empiric-mathematical method: von Meier et al. took the second derivative of the pavement profile of no more than $d^*$, a parameter reflecting the tire hardness, as the tire deformation line [25]. In the ROSANNE project, the area ($S$) enclosed by a baseline and the pavement profile above the baseline was controlled, and the part of the pavement profile was taken as the enveloped pavement—indentor method [3]. Andersen [26] proposed a method similar to the indentor method but took the bearing area ratio as the control parameter. The values of the control parameters in these methods depend on the experimental test, which should be close to the real contact since the parameters are the calculation or iteration termination conditions for the enveloping profile. However, the method by von Meier recommends $d^*$ values based on the tire deformation on the surface made by steel spheres with different radii and distribution distances. The $S$ value in the indentor method was recommended by the deformation of plasticine in the 'pavement' formed by long wooden strips with triangular sections of different spacing under the rolling tires.

In general, determining the depth of rubber penetrated in the pavement becomes necessary due to the incomplete contact between the tire and pavement with multiscale surface roughness. However, most of the existing methods for penetration depth rely on exact theoretical analysis with several conditional assumptions, or on the tests with complicated testing processes and less time efficiency, or they cannot reflect the distribution characteristics of the pavement texture. To this end, this study intends to propose a new method that can reflect the pavement morphology and the conditions under which the contact occurred, to test the rubber penetration depth on the pavement. The content includes:

- introducing the measurement principle and discussing its feasibility;
- analyzing the affecting factors for the new methods; and
- evaluating its application by comparing it with the latest methods.

## 2. Materials and Methods

### 2.1. Measurement Principle

The new procedure in this study relies on the rubber-pavement contact area and the bearing area curve (BAC) of the pavement (Figure 2a). The BAC is also called the areal material ratio curve [27]. It characterizes the variation in sectional area as one moves from the top to the bottom of the pavement surface. Figure 2a shows the accumulative distribution of the areal ratio of the material bearing the load on the surface from the highest point down. Every rough surface has a certain BAC, determined by its morphology.

Then, suppose the projected areal ratio of the contact area between the tire and pavement q = S2/S1 (Figure 2b) can be measured. In that case, the rubber penetration depth ($d$) can be determined according to the BAC. In Figure 2b, S1 is the nominal contact area between the tire and pavement, which is in the same order of size as that of a tire-pavement interface, while S2 is the projection of the part of the particle enveloped by tread rubber onto the horizontal plane. From here onwards, the new method proposed in this study is referred to as the S-BAC method.

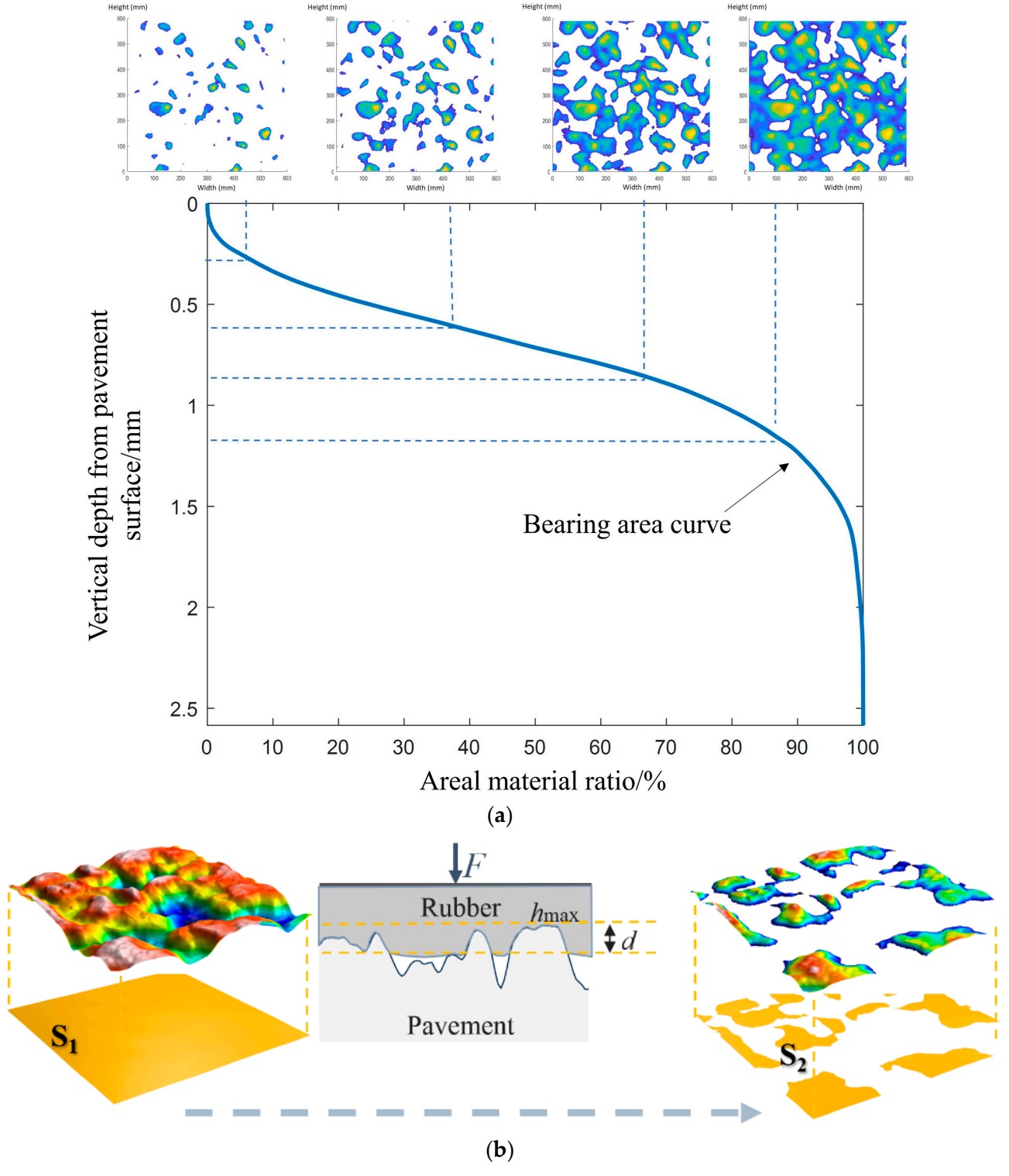

**Figure 2.** Schematic representation of the S-BAC method for obtaining the rubber penetration depth. (**a**) Bearing area curve of a pavement surface; (**b**) Projection of nominal and actual contact areas onto the horizontal plane.

*2.2. Test Objects*

2.2.1. Rubber Block Representing Tire Tread

The contact between the tire and pavement is determined by many factors, such as load, pavement texture, tire inflation, tire tread pattern, tire age, and tread rubber hardness [28]. Furthermore, the vehicle operation, such as acceleration, deceleration,

turning, and road alignment (e.g., vertical and horizontal slope), could significantly affect the pressure distribution and contact. This study simplified the contact between the tire and pavement as a static contact between a rubber plate and the pavement specimens under a vertical load.

There are various tread patterns, which will cause different geometric stiffness properties of tread and change the rubber penetration depth [29]. Thus, it is virtually impossible to designate a reference tire with a reasonably representative tread pattern. The simplification can better highlight the effect of different pavement textures on the rubber penetration depth. Selecting a smooth rubber block with the same hardness as the tire tread rubber is crucial, as hardness determines tread deformation under load and is closely related to the type and distribution of tread pattern.

### 2.2.2. Specimen Preparation

In principle, when using the S-BAC method to obtain the penetration depth between a rubber block and the pavement, the pavement specimens can be obtained by various methods, such as directly drilling from the road, molding indoors, or copying pavement textures.

To not damage the road and fully represent the pavement texture, this study recorded the surface texture by a 3D laser scanner in the field (Step 1 in Figure 3). Then, specimens with the texture of the pavement were replicated in light-colored (white) resin materials using 3D printing technology (Step 3 in Figure 3). The reproduced specimen covers a length and a width of 60 mm, which is sufficient for enveloping.

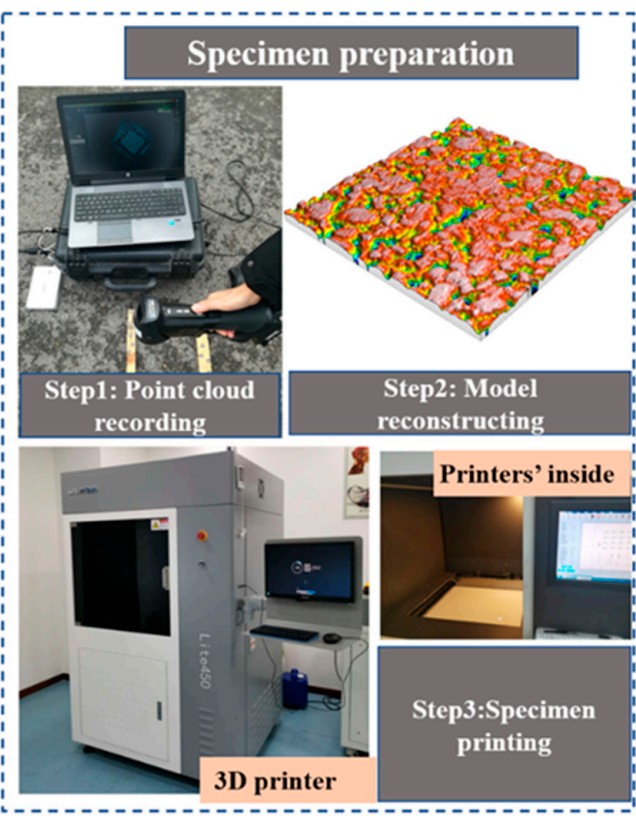

**Figure 3.** Production procedure for the specimens.

Previous studies and experience have suggested that texture wavelengths in the range of 5~50 mm contribute more to the rubber penetration depth than shorter texture wavelengths [30]. Mogrovejo et al. [31] and Goubert and Sandberg [3] used a sampling spacing of 0.2 and 0.5 mm, respectively, to obtain the pavement profile and the penetration depth for a tire on the pavement. Thus, the HandyScan 3D scanner (Creaform™, Lévis, QC, Canada), which can provide a resolution of 0.1 mm, an accuracy of 0.04 mm, and the

sampling area is not limited, was used to obtain the pavement texture [32]. An industrial 3D printer lite600 (UnionTech[TM], Shanghai, China) based on stereolithography, with a high printing resolution of 0.1 mm and 0.05 mm in horizontal and vertical directions, was used to minimize the loss of captured texture details. Moreover, the elastic modulus of the resin materials used for 3D printing was comparable to that of typical asphalt pavements (about 3000 MPa at 25 °C) after curing.

In this way, both the texture and hardness of the printed specimens could reasonably well realistically represent the pavement surface, at least with relevance to the rubber penetration depth. Additionally, the elastic modulus of the printed specimens was much higher than that of rubber (about 8 MPa at 25 °C). The specimen's compressive deformation and its effect on the contact patches could be ignored.

### 2.3. Contact Area Measurement

### 2.3.1. Rubber/Pavement Contact Area Marked Using the Staining Method

Instruments that can precisely provide specific loads are essential for this purpose. This study used the universal test machine (UTM) (IPC[TM], Liscate, Italy), which can precisely control the temperature (resolution 0.1 °C) and load (±1 N). A self-designed fixture was used with the UTM (Figure 4). A rubber block chosen to represent the tire tread, the bottom of which was stained, was glued to the underside of the steel plate of the fixture.

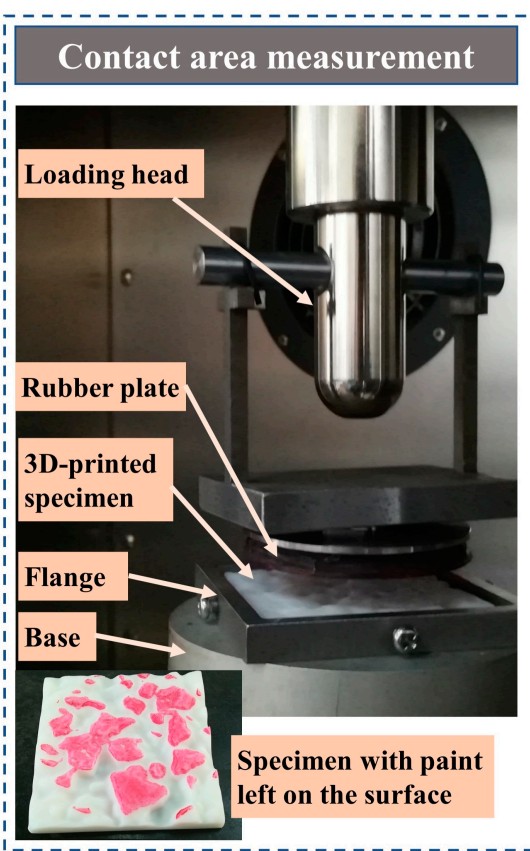

**Figure 4.** Loading device, self-designed fixture, and a specimen with paint on the surface.

During the test, the specimen was fixed on the base using a flange, and the base was accurately aligned with the loading head (Figure 4). The UTM works according to set processes, at which the load, load duration, and loading waveshape can be designed in advance. In this study, the contact time was 10 s, during which the load was constant. After squeezing the rubber against the specimen, the stained area on the specimen surface marked the contact area (Figure 4). From now onwards, the contact area measurement is referred to as 'the staining method'.

A potential issue with staining is that some excessive paint may be pressed somewhat outside of each local contact area, thus blurring the edges and overestimating the contact area. The paint, in this case, had an estimated average thickness of only 9~20 μm, leaving little paint available to be squeezed outside the contact.

### 2.3.2. Contact Area Ratio

According to Section 2.1, the proportion of the real contact area to the nominal contact area needs to be considered to determine the tire penetration depth using the S-BAC method. This study used scanning equipment, such as a printer with a scan function, to record the image of the stained specimens' surfaces (Figure 5), and the region of interest can be obtained using a clipping tool. Then, image processing was used to obtain the corresponding binary image and accurately calculate the contact area ratio.

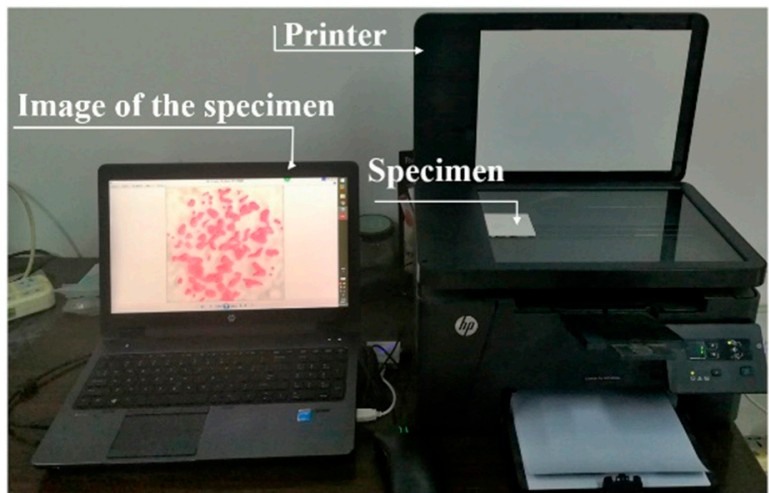

**Figure 5.** Obtaining the image of the specimens with paint left on the surfaces.

This study developed an algorithm (Method 1) in MATLAB based on the Lab color space and compared it with the conventional gray threshold segmentation method (Method 2) [33]. Interpretations using the two methods are shown in Figure 6.

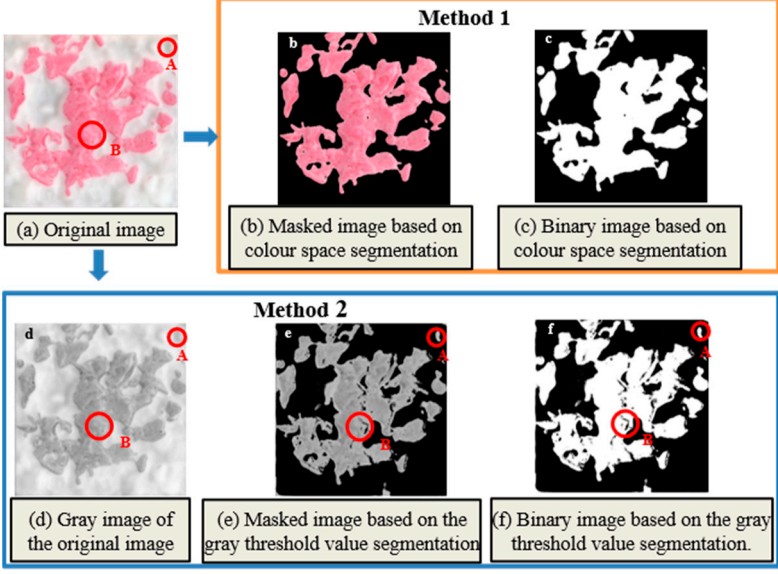

**Figure 6.** Comparison between the two area computation methods.

The contact area recognition of Method 1 compares the chromatic components of each pixel with pre-set thresholds to classify pixels into target and background classes (Figure 6b). In contrast, Method 2 first converts the original image (Figure 6a) into a gray image (Figure 6d). The pre-set thresholds for both Methods 1 and 2 are determined using the K-means clustering method. Method 2 could incorrectly interpret the non-contact area as a contact area (e.g., region A) or misinterpret the contact area as a non-contact area (e.g., small 'cracks' in region B). These errors could be attributed to the limitation of gray threshold segmentation in effective segment regions with similar gray levels but different colors (Figure 6e). Notably, in images with distinct color differences, the contact area in red can be identified more accurately. The ratio of real contact area to nominal contact area can be calculated using the following equation after producing the binary images.

$$\text{Contact area ratio } (\%) = \frac{Num_{pixel=1}}{Num_{total}} \times 100 \tag{1}$$

Here, $Num_{pixel=1}$ is the number of pixels with a pixel value 1, $Num_{total}$ is the total number of pixels in the nominal contact area.

### 2.3.3. Resolution of the Staining Method

Previous studies have indicated that the observation scale significantly affected the observed contact area between two rough surfaces [34]. When observed at a larger scale (i.e., poorer resolution, ignoring the finer parts of the macrotexture), the observed area becomes larger than the true area. The pavement texture consists of roughness that extends across several scales, which makes the contact area dependent on the observed scale.

The staining method introduced in Section 2.3.1 regards the area with paint as the contact area. When the rubber block is squeezed onto surfaces with a finer texture by a load above a certain threshold, the stain may be squeezed to fill the space between the contact regions, and the detected contact area would no longer decrease. In other words, the staining method cannot help identify the contact area if it is pictured with a too high resolution.

Several artificial surfaces were produced with different 'smoothness' to investigate the resolution at which the contact can be detected using the staining method. The different smoothness was obtained by applying low-pass filters to the original artificial surface. The artificial surface was generated by designing a Gaussian height probability distribution based on the power spectral density (PSD) curve, as shown in Figure 7 [35]. The shape of the PSD curve was determined by the fractal dimension (*H*), the root-mean-square height of the surface (*Sq*), and the nominal maximum aggregate size (*NMAS*) of the asphalt pavement [36]. Based on measurements of these factors from the real pavement surface, this study set the *Sq* as 0.5 mm, the *H* as 2.2, and the *NMAS* as 12.6 mm. The low-pass filtering cut-off, and also the minimum texture wavelength (λs) on a surface, ranged from 12.6 mm down to 0.2 mm. Nine digital surface variants with different λs were then produced by a 3D printer as physical samples representing surfaces at different observation resolutions (Figure 8). These physical samples were used as specimens in the setup shown in Figure 4 to determine the rubber penetration depth.

Applying the staining method to the artificial surfaces, the images in Figure 8 were obtained, where the number in each image represents the λs of the surface. The reddish region shows the contact area between the specimen and the flat rubber block for a load of 50 N. The change in the red area on the surface from picture to picture shows the effect of the observation resolution on the contact area. For the larger λs printed surface, the observed contact area is relatively high. When λs decreases, the contact area gradually shrinks, and the contact spots get separated. At the lower λs, the contact area becomes relatively independent of the λs.

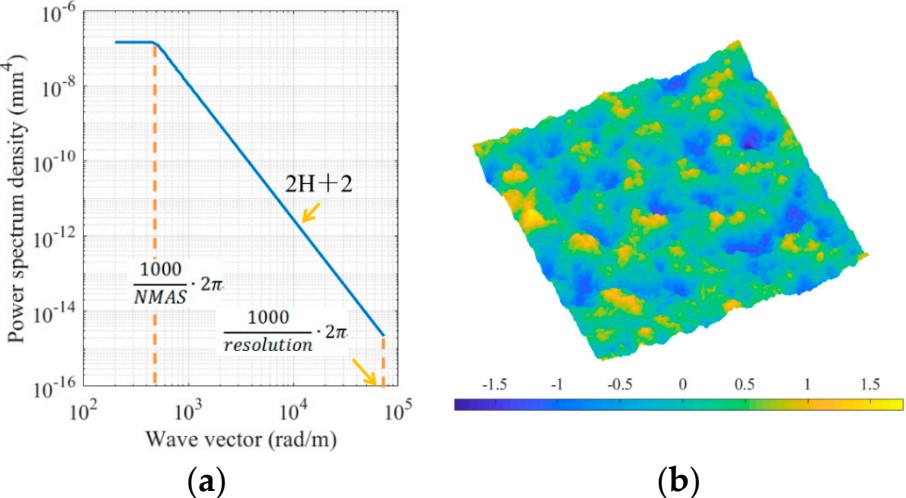

**Figure 7.** Generating a surface with Gaussian height probability distribution based on the power spectral density. (**a**) power spectrum density curve; (**b**) artificially designed surface.

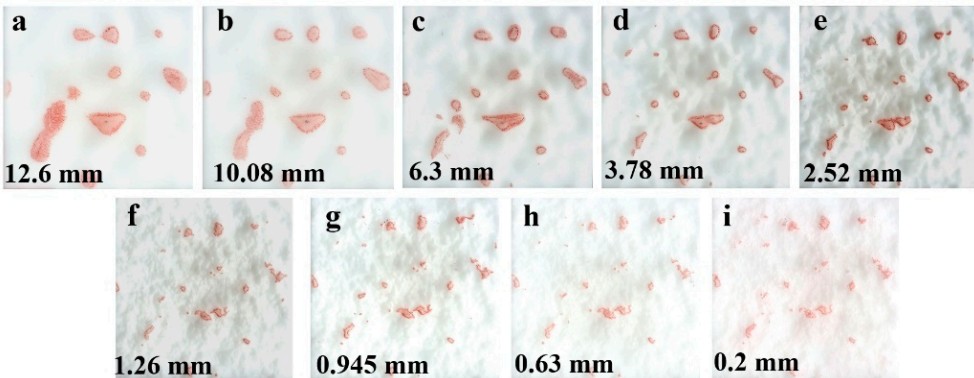

**Figure 8.** Contact area on artificially designed surfaces with different minimum texture wavelengths (50 N).

Figure 9 shows the contact areas obtained on surfaces with different λs under different loads and rubber hardness conditions. The rubber hardness of the tested rubber blocks included 55 and 68 Shore A, measured according to GB/T 531.1 [37]. The applied forces were 50, 500, 1000, 1500, and 2000 N. The x-axis refers to the λs applied to each surface. The y-axis shows the measured contact area ratio, which is the proportion of the contact area to the area of the rubber bottom. Figure 9 shows that when λs begins to decrease, the contact area ratio rapidly decreases. However, the paint, when squeezed by the rubber, can fill the emerging fine texture on the surface. The detected contact-area ratio does not decrease when the roughness provided is fine enough. For the artificially designed surfaces with Gaussian height probability distributions, the texture wavelength below which the ratio starts to become constant appears at 0.6 mm.

Compared to the artificially designed surfaces, the profile of the top part of the real pavement surface is flatter, which means that the pressure distribution at the interface between the rubber and pavement would be more uniform. However, using test surfaces with such fine-textured peaks as were generated in this study makes it possible to determine the proper acquired resolution.

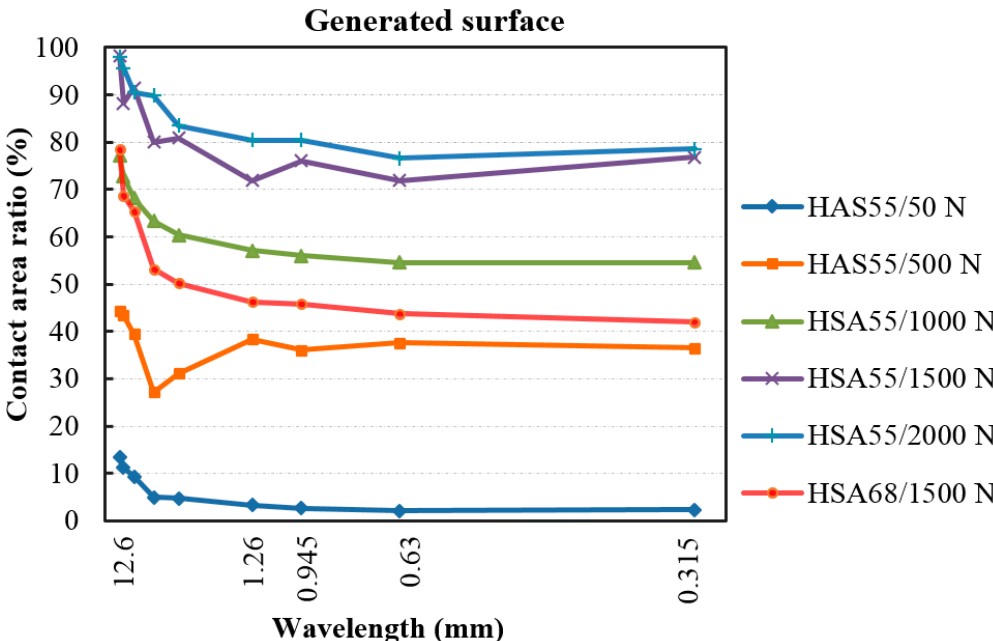

**Figure 9.** Contact areas determined for surfaces with different minimum texture wavelengths under different loads and rubber hardness conditions.

Nevertheless, the microtexture of the real pavement, with dimensions comparable to the thickness of the paint, would be rougher than on the generated and 3D-printed surfaces. Therefore, the paint is less likely to be squeezed out of the contact area on the pavement and thus avoids showing larger contact areas than the actual ones.

This study conservatively estimates that the determination of the rubber-pavement contact area, which can be estimated using the staining method, requires a resolution in texture wavelength of approximately 0.6 mm.

### 2.4. Bearing Area Curve and Analysis Scale

The bearing area curves (BACs) can be obtained using 3D surface texture data. A smaller sampling interval can capture more surface details than long ones, making the measured data more realistic. However, the analysis in Section 2.3.3 shows that the staining method could not detect the contact area for texture wavelengths shorter than 0.6 mm. According to Section 2.1, the BAC should be determined with the same resolution as the contact area. Generally, most scanners (or other texture-recording devices) operate with 0.1 mm or higher resolutions. Hence, ideally, the captured texture wavelengths are longer than 0.2 mm. Figure 9 shows that it may not be necessary to filter the surface data to remove the roughness with shorter texture wavelengths than around 0.6 mm as the contact areas do not depend on these wavelengths.

Figure 10 shows the BACs calculated from artificial surfaces with different 'smoothness' by applying low-pass filtering with cut-offs at texture wavelengths of 0.2, 0.5, 0.63, 0.945, 1.26, and 2.5 mm (using a Gaussian filter response). After filtering (removing) the short waves, it appears in Figure 10 that the top part (higher height) accounts for a higher ratio of the surfaces' areal material; in other words, the texture decreases.

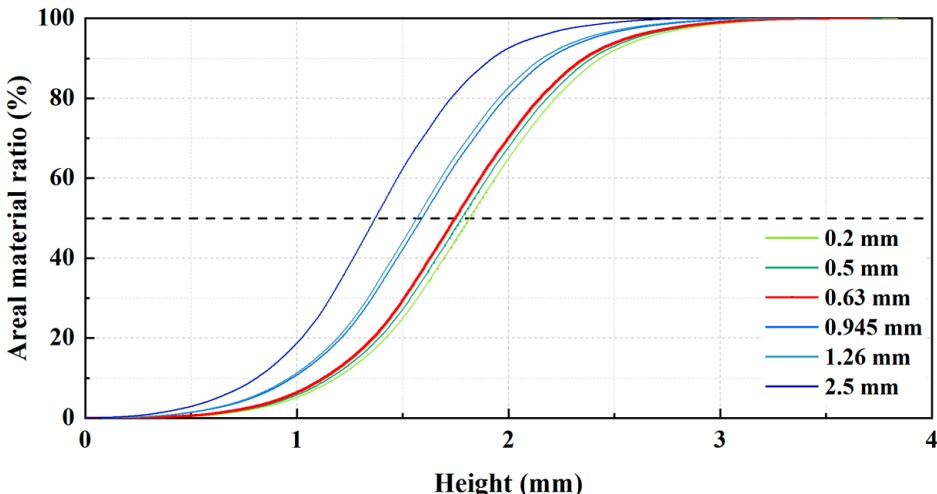

**Figure 10.** Bearing area curves calculated for artificially designed surfaces with texture wavelengths cut-off at 2.5 to 0.2 mm.

This study uses the contact area ratio measured with the staining method and the BAC of the specimen's surface data to obtain the penetration depth of rubber on pavements. The analysis in Section 2.3.3 suggested that the measured contact area ratio was most appropriate for a resolution of about 0.6 mm, expressed as the texture wavelength. Therefore, the resulting rubber penetration depth should be larger than the true value if the surface data used to calculate the BAC contains texture wavelengths below 0.6 mm. Specifically, when the areal material ratio is 50%, the penetration depths obtained for surfaces with λs of 0.2, 0.5, and 0.63 mm are 1.816, 1.781, and 1.751 mm, respectively. Consequently, for an artificial surface with Gaussian height probability distribution, the error caused by the analysis resolution of the BAC was less than approximately + 5%, based on a reference with a minimum texture wavelength of 0.2 mm.

For a realistic asphalt pavement, the surface generally has a negative skewness, and the top part of the profile is relatively flat due to the rolling process during construction and the accumulated vehicle load, as illustrated in Figure 11. The numbers in brackets indicate the λs of the surfaces. The upper profile curve represents the case of the pavement, and the lower one represents the case of the artificial surface with a Gaussian height probability distribution.

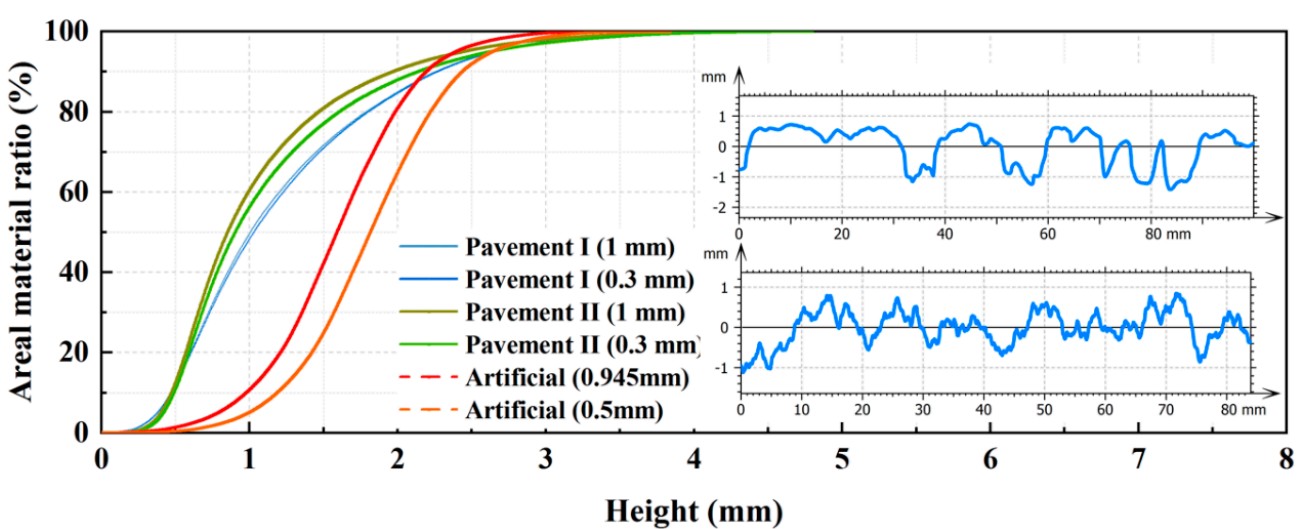

**Figure 11.** Bearing area curves and profiles obtained from the artificial surface and two pavements, respectively.

According to the BACs, the analysis scale of texture wavelength makes a smaller difference for the pavement surfaces than for the artificial surfaces. The profiles in blue (Figure 11) show that the top part of the pavement is flatter than on the artificial surface. In addition, the texture at the top part of the pavement contributes less to the surface height than the middle and bottom parts do. Therefore, when the staining method from Section 2.3 is used to determine the contact area, the error in the penetration depth obtained using the BAC is below 5%, even without filtering the texture with a texture wavelength below 0.6 mm.

## 3. Comparison of Different Methods to Determine Rubber Penetration Depth

### 3.1. Test Objects for the Comparison

This section compares the penetration depth obtained using the S-BAC, indentor method, and the enveloping method proposed by von Meier et al. [25]. This study compares the results by applying the three methods on the same surfaces. The surface texture data were captured on: (1) 29 asphalt pavement sections, the mean texture depth (MTD) of which ranges from 0.31 to 1.41 mm, and (2) 15 artificial surfaces designed by a similar method as described in Section 2.3. The texture on these surfaces was captured and calculated by the HandyScan 3D scanner (Creaform$^{TM}$, Lévis, QC, Canada).

### 3.2. Application of the S-BAC Method

The procedure to determine the rubber penetration depth for each sample using the S-BAC method is shown in Figure 12. The detailed information is as follows:

1.  Prepare the specimens by the method described in Section 2.2.2. The size of specimens should be sufficient for enveloping. This study tests the $60 \times 60$ mm$^2$ specimens produced by a 3D printer.
2.  The fixture described in Section 2.3.1 is attached to the loading device (e.g., the UTM). Then, the specimen and the fixture are centered. The loading head is raised to a suitable height (e.g., 15 mm above the specimen), and the rubber block's bottom is painted.
3.  The loading process is set up and started using the user interface. After loading, the head is raised, and the specimen with the paint on the surface is dried.
4.  The contact area ratio is determined using the method described in Section 2.3.2.
5.  The surface morphology of the specimen is scanned, and the accumulative height probability distribution (BAC) of the specimen is determined.
6.  According to the test principles in Section 2.1, the rubber penetration depth is obtained using the contact area ratio obtained in Step 4 and the BAC obtained in Step 5.
7.  The same procedure is repeated for each of the 44 samples introduced in Section 3.1.

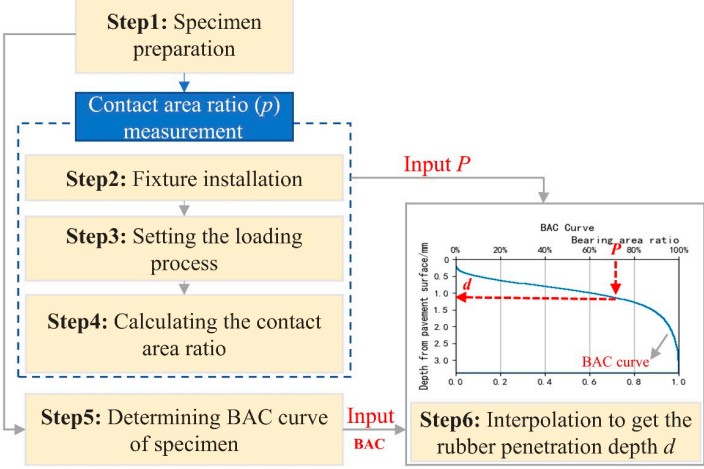

**Figure 12.** Procedure to determine the rubber penetration depth.

The vertical loads for the (S-BAC method) measurements were 50, 500, 1000, 1500, and 2000 N, and the corresponding pressures were 0.018, 0.18, 0.35, 0.53, and 0.71 MPa. Two rubber blocks were used in this study, with a hardness of 55 and 68 Shore A, respectively, 6 mm thick and 60 mm in diameter.

### 3.3. S-BAC Method and Enveloping Method by von Meier et al. [25]

Figure 13 shows the enveloping profiles obtained using the method by von Meier et al. [25], where the parameter d* depended on the mechanical characteristics of the tire. The value $d* = 0.054$ mm$^{-1}$, derived from a measurement of the deformation of a tire pressed onto various idealized profiles, was suggested [25].

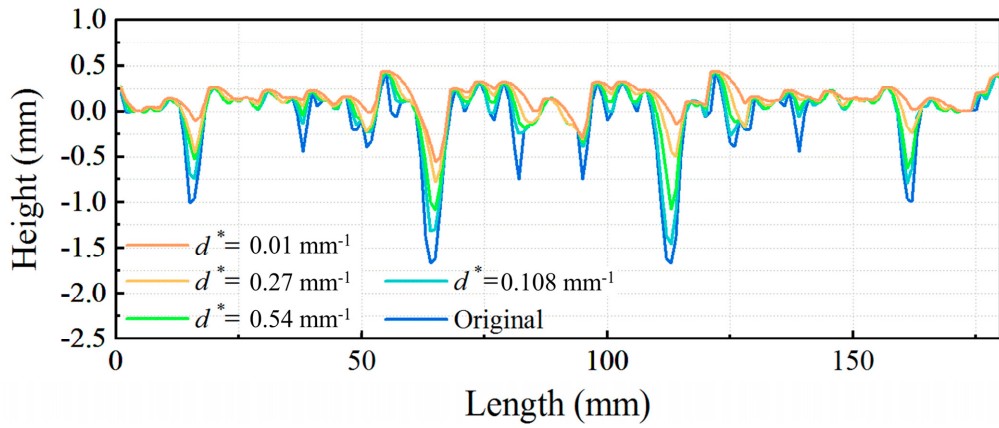

**Figure 13.** Effect of the *d** on enveloping profiles obtained using the method by von Meier et al. [25].

Figure 13 shows that the penetration depth of the tire into the pavement texture was greatly affected by the width between two adjacent asperities in the pavement profile. This study used a baseline with a length of 100 mm to calculate the enveloping profiles. Each profile was divided into two sections. The difference between the average of the highest and lowest points for these two sections was used as the rubber penetration depth (Figure 14). This could reduce the effect of space between adjacent asperities. The results for 10 profiles for each surface were taken as the representative value.

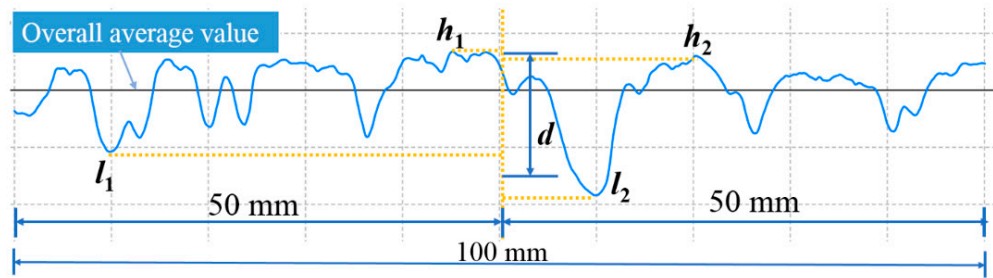

**Figure 14.** Schematic of the terms baseline, and the penetration depth obtained using the method by von Meier et al. [25].

Figure 15 shows the penetration depth obtained using the S-BAC method and the method by von Meier et al. [25]. The penetration depth obtained using the two methods is identical for the solid gray line ($y = x$). The solid red line refers to the regression line between the two variables. For 44 samples, when Pearson's r (correlation coefficient) is higher than 0.29 ($\alpha = 0.05$), there is a significant correlation between the two variables. There is a strong correlation between the penetration depth obtained by the S-BAC method and the von Meier method, except for the HSA68/1500 N condition.

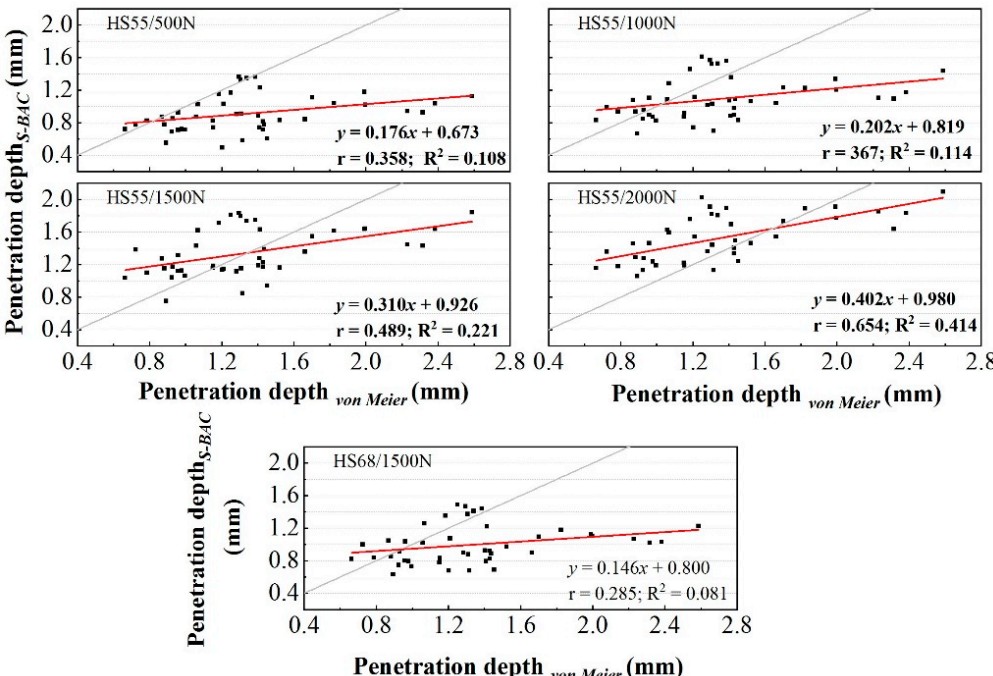

**Figure 15.** Comparison of penetration depth between the S-BAC and von Meier method.

Most penetration depths for the 44 surfaces obtained by the S-BAC method are below that obtained by the von Meier method for low-pressure conditions. The two penetration depths are equal on surfaces with small penetration depths obtained using the von Meier method. As the pressure increases or the rubber hardness lowers, the tail of the red line gradually rises.

### 3.4. S-BAC Method and the Indentor Method

Figure 16 shows the enveloping profiles obtained by the indentor method with different S. The value of S was recommended as 6 or 10 mm$^2$ by the ROSANNE project [3]. In this study, the height difference between the horizontal line's height at which a specific S can be satisfied and the pavement profile's highest point refers to the penetration depth obtained by the indentor method. The penetration depth can be calculated by iteration against *S*, and the error is 0.01 mm$^2$. The larger *S* results in a higher value of penetration depth (Figure 16).

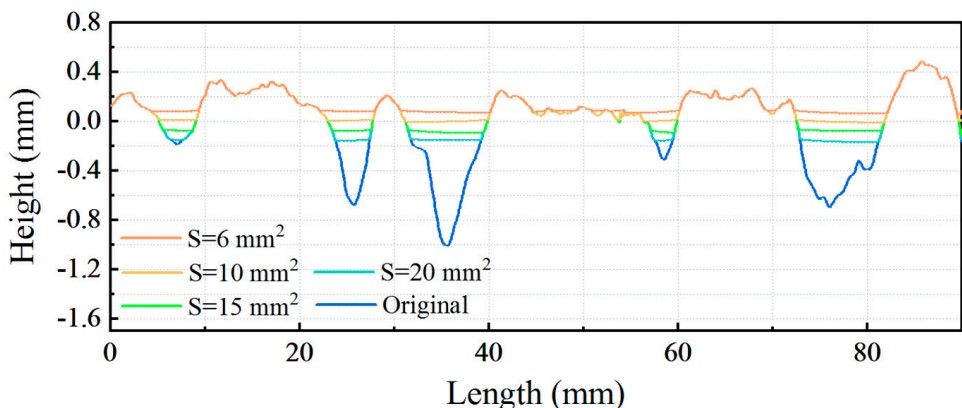

**Figure 16.** Tire-road enveloping profile obtained by the indentor method with different S.

Figure 17 compares the penetration depth obtained by the S-BAC and indentor methods. The depth obtained by the indentor method is the average of the values calculated from 10 profiles extracted from the specimens' surfaces.

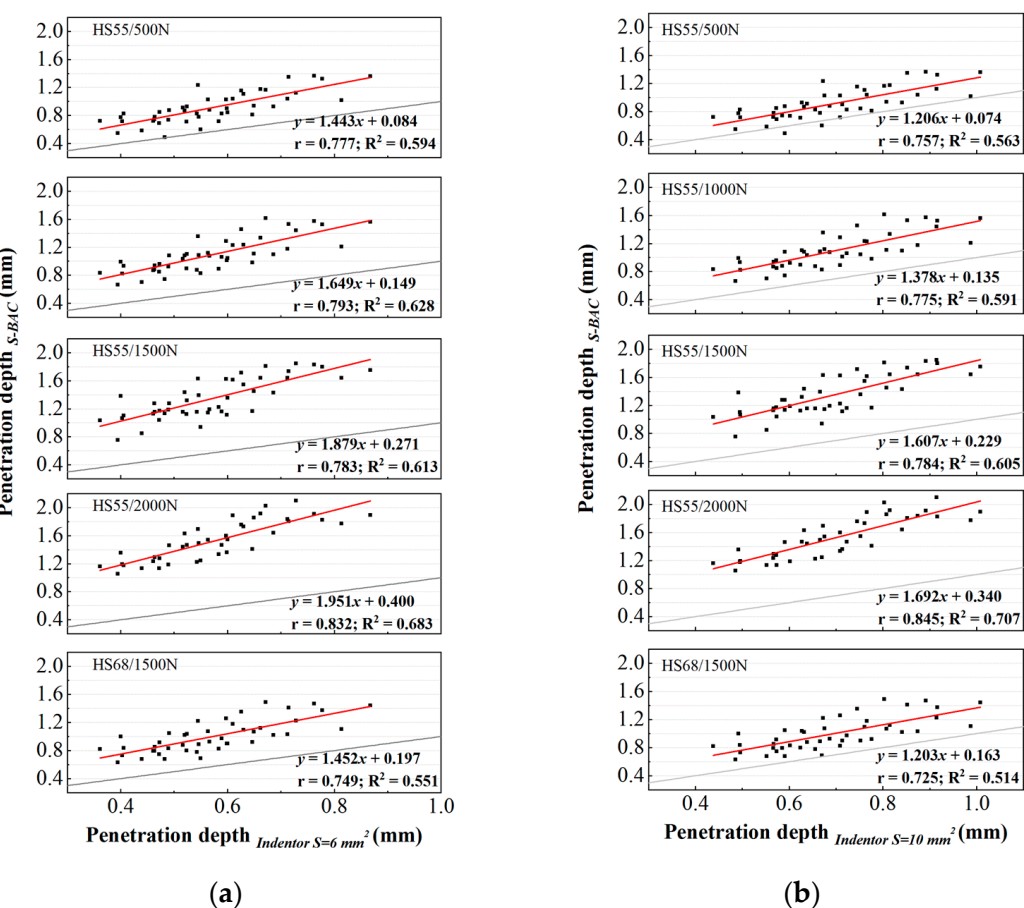

**Figure 17.** Relationship between rubber penetration depth obtained using the S-BAC and indentor method. (**a**) indentor $S = 6$ mm$^2$; (**b**) indentor $S = 10$ mm$^2$. The gray lines are the isoclinic line, the red lines are fitted lines.

Figure 17 reveals a close linear relationship between the penetration depth obtained by the S-BAC and indentor methods. The penetration depths obtained by the former are always higher than the latter under the test conditions in this study. More specifically, the value of depth obtained using the S-BAC method is about 1.2~3 times larger than for the indentor method. The multiple gradually increased with increasing pressure or decreasing rubber hardness.

## 4. Application of Rubber Penetration Depth Obtained from Different Methods

### 4.1. Friction Coefficient and Pavement Texture Measurement

This study chose 23 asphalt concrete test sections in Xi'an, China. The pavement texture was recorded using the HandyScan 3D scanner introduced in Section 2.2.2. The friction coefficient on these sections was measured using the T2GO shown in Figure 18 (SARSYS-ASFT$^{TM}$, Köpingebro, Sweden). The device creates a fixed slip of 20%, which effectively simulates the braking action.

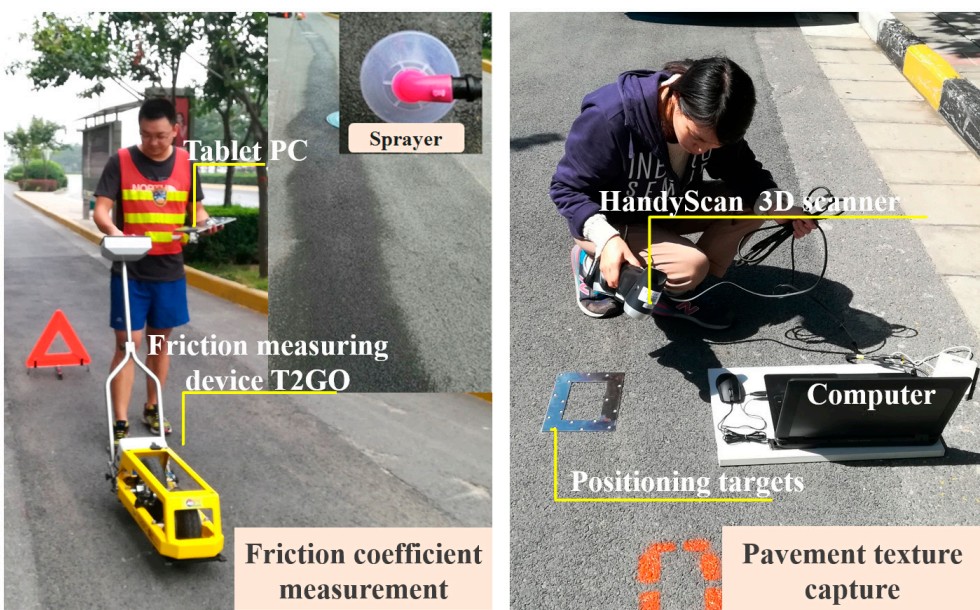

**Figure 18.** Measuring the low-speed friction coefficient and capturing the pavement texture.

The friction recorded speed during the test was 2~3 km/h, and the measurement length on each test section was about 15 m. Each test was repeated twice. The measurement interval for friction was set to 30 mm (0.1 feet) with the interface controls located at the top of the handlebar. Based on the walking speed during sprinkling (1.5~2 km/h), the sprinkling width (0.3 m), the water outflow speed (15~25 mL/s), and ignoring evaporation, the water film thickness was estimated at approximately 0.1~0.2 mm. Following the standard correction factor of 0.003 per 1 °C, the friction measurements obtained using the T2GO device (SARSYS-ASFT$^{TM}$, Köpingebro, Sweden) were adjusted to a reference temperature of 20 °C [38]. The sampling interval for the pavement texture was 0.2 mm, and the scanning area was $400 \times 280$ mm$^2$. Three positions, evenly distributed along the test path, were scanned in each section.

Due to the limitation of scanning instruments, only pavement macrotexture was captured in the field. However, microtexture significantly affects the friction coefficient between the rubber and the pavement [39]. This study took two measurements to equate the effect of the microtexture on the measured friction coefficient. First, we wet the pavement before measuring the friction coefficient; second, we chose test sites with a slight difference in microtexture. In this case, the water occupies part of the microtexture on the pavement. Furthermore, most of the test sections we chose were on the right side of the right-most lane on the road, where the aggregates suffered little abrasion, and the microtexture was still almost covered by asphalt.

Figure 19 shows the measured friction coefficient. The water film thickness during friction measurement was 0.1~0.2 mm, so there was sufficient time for a low-speed test to eliminate the water between the tire and the pavement surface. The nominal pressure between the friction-measuring device and the pavement was 0.52 MPa, and the rubber hardness was about 68 HSA [40]. Therefore, the depth of the T2GO's tire penetration in the pavement was tested using the S-BAC method at 0.52 MPa, 68HSA, under dry conditions. Table 1 shows the tire penetration depth on the pavement.

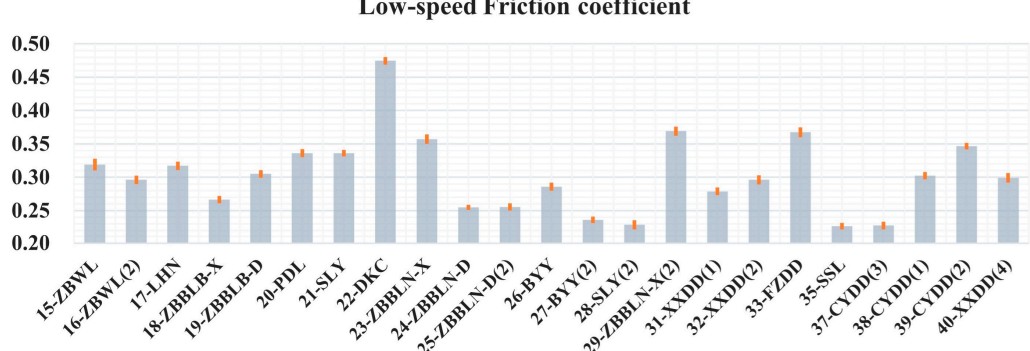

**Figure 19.** Measured friction coefficient by T2GO device.

**Table 1.** Rubber penetration depth obtained by different methods in mm.

| Test Sections | S-BAC Method | | Indentor Method | | | | von Meier $d^* = 0.054$ mm$^{-1}$ | |
| | | | $S = 6$ mm$^2$ | | $S = 10$ mm$^2$ | | | |
| | Depth | Stdev | Depth | Stdev | Depth | Stdev | Depth | Stdev |
|---|---|---|---|---|---|---|---|---|
| 15-ZBWL | 0.84 | 0.05 | 0.49 | 0.04 | 0.60 | 0.05 | 1.12 | 0.32 |
| 16-ZBWL(2) | 1.12 | 0.02 | 0.68 | 0.01 | 0.84 | 0.01 | 2.08 | 0.16 |
| 17-LHN | 1.04 | 0.12 | 0.56 | 0.01 | 0.69 | 0.01 | 1.60 | 0.10 |
| 18-ZBBLB-X | 1.11 | 0.13 | 0.62 | 0.05 | 0.78 | 0.08 | 1.89 | 0.32 |
| 19-ZBBLB-D | 1.17 | 0.06 | 0.64 | 0.02 | 0.79 | 0.03 | 1.88 | 0.08 |
| 20-PDL | 0.86 | 0.03 | 0.48 | 0.01 | 0.58 | 0.02 | 1.03 | 0.07 |
| 21-SLY | 1.06 | 0.06 | 0.48 | 0.01 | 0.58 | 0.02 | 1.03 | 0.07 |
| 22-DKC | 0.78 | 0.03 | 0.46 | 0.02 | 0.56 | 0.02 | 1.06 | 0.05 |
| 23-ZBBLN-X | 1.12 | 0.12 | 0.62 | 0.04 | 0.76 | 0.06 | 1.48 | 0.14 |
| 24-ZBBLN-D | 1.24 | 0.16 | 0.68 | 0.08 | 0.84 | 0.09 | 1.79 | 0.07 |
| 25-ZBBLN-D(2) | 0.74 | 0.02 | 0.44 | 0.01 | 0.53 | 0.01 | 0.99 | 0.11 |
| 26-BYY | 1.08 | 0.20 | 0.62 | 0.05 | 0.75 | 0.06 | 1.65 | 0.37 |
| 27-BYY(2) | 1.15 | 0.01 | 0.63 | 0.04 | 0.76 | 0.05 | 1.60 | 0.15 |
| 28-SLY(2) | 1.40 | 0.16 | 0.77 | 0.08 | 0.96 | 0.09 | 2.75 | 0.60 |
| 29-ZBBLN-X(2) | 0.76 | 0.04 | 0.48 | 0.04 | 0.59 | 0.05 | 1.03 | 0.35 |
| 31-XXDD(1) | 1.02 | 0.03 | 0.62 | 0.03 | 0.77 | 0.03 | 1.87 | 0.13 |
| 32-XXDD(2) | 0.89 | 0.03 | 0.53 | 0.04 | 0.66 | 0.05 | 1.41 | 0.39 |
| 33-FZDD | 0.84 | 0.04 | 0.50 | 0.03 | 0.62 | 0.05 | 1.12 | 0.17 |
| 35-SSL | 0.99 | 0.05 | 0.60 | 0.02 | 0.73 | 0.04 | 1.50 | 0.06 |
| 37-CYDD(3) | 0.81 | 0.14 | 0.44 | 0.08 | 0.53 | 0.09 | 0.72 | 0.13 |
| 38-CYDD(1) | 0.86 | 0.16 | 0.48 | 0.07 | 0.58 | 0.08 | 1.05 | 0.27 |
| 39-CYDD(2) | 0.78 | 0.07 | 0.45 | 0.03 | 0.55 | 0.04 | 0.92 | 0.22 |
| 40-XXDD(4) | 0.89 | 0.17 | 0.55 | 0.09 | 0.67 | 0.10 | 1.27 | 0.12 |

### 4.2. Effect of Rubber Penetration Depth on Pavement Texture-Friction Relationship

After obtaining the penetration depth, the texture parameters were calculated based on cloud data above the depth. Two commonly used parameters (root-mean-square surface height $S$q and root-mean-square slope of the surface $S$dq), considered to significantly affect the actual contact area and pressure distribution, which plays a pivotal role in determining the friction coefficient formation, were selected [41]. The calculation of these two parameters required Equations (2) and (3). The average of the parameters at three positions on the test path was used to represent each test section.

$$S\text{q} = \sqrt{\frac{1}{A}\iint\limits_{A} z^2(x,y)dxdy} \tag{2}$$

$$Sdq = \sqrt{\frac{1}{A}\iint\limits_{A}\left[\left(\frac{\partial z(x,y)}{\partial x}\right)^2 + \left(\frac{\partial z(x,y)}{\partial y}\right)^2\right]dxdy} \tag{3}$$

Here, $z(x, y)$ is the surface height measured from the average plane with $z = 0$, A is the evaluation area.

Figure 20 shows the effect of the penetration depth obtained by different methods on the relationship between the pavement texture parameters and friction coefficient. The parameters $Sq$ and $Sdq$ calculated based on different penetration depths are shown at the horizontal axes of Figure 20a,b, respectively. The low-speed friction coefficient is at the vertical axes. The red zones are shown at the 95% confidence band to compare these equations' prediction accuracy, and the smaller confidence interval indicates the more accurate prediction. Table 2 shows the details of the equations in Figure 20.

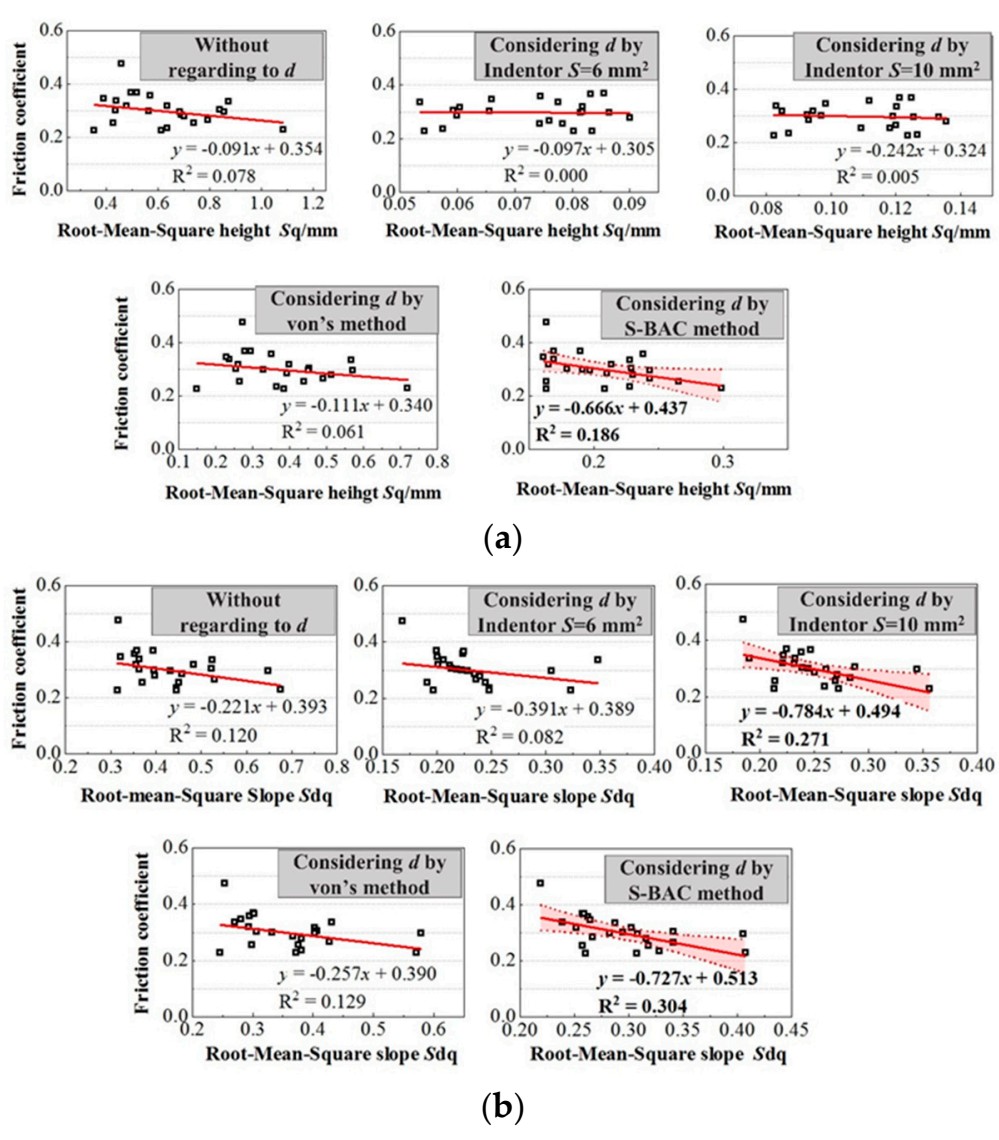

**Figure 20.** Effect of penetration depth on the relationship between texture parameters and the friction coefficient. (**a**) Texture parameter $Sq$ and the friction coefficient; (**b**) Texture parameter $Sdq$ and the friction coefficient.

**Table 2.** Parameters of the regression lines and the hypothesis testing.

| Parameter | Method | Equation | a | b | Pearson's r | R$^2$ | F Test -$p$ |
|---|---|---|---|---|---|---|---|
| RMS height of the surface $Sq$ | Full depth | | −0.091 | 0.354 | −0.279 | 0.078 | 0.197 |
| | S-BAC | | −0.666 | 0.437 | −0.432 | 0.186 | 0.040 * |
| | Indentor $S$ = 6 mm$^2$ | | −0.097 | 0.305 | −0.018 | 0.000 | 0.935 |
| | Indentor $S$ = 10 mm$^2$ | | −0.242 | 0.324 | −0.068 | 0.005 | 0.758 |
| | Von Meier | $y = a \times x + b$ | −0.111 | 0.340 | −0.246 | 0.061 | 0.258 |
| RMS slope of the surface $Sq$ | Full depth | | −0.221 | 0.393 | −0.346 | 0.120 | 0.105 |
| | S-BAC | | −0.727 | 0.513 | −0.551 | 0.304 | 0.006 * |
| | Indentor $S$ = 6 mm$^2$ | | −0.391 | 0.389 | −0.287 | 0.082 | 0.183 |
| | Indentor $S$ = 10 mm$^2$ | | −0.784 | 0.494 | −0.521 | 0.271 | 0.011 * |
| | Von Meier | | −0.257 | 0.390 | −0.360 | 0.129 | 0.092 |

*: being significant with a 95% confidence.

The R2 and confidence intervals indicate that the relationship between the texture parameters ($Sq$, $Sdq$) and the friction coefficient is statistically significant when calculating the penetration depth based on the S-BAC method. Though $Sq$ and $Sdq$ have a weak relationship with the low-speed friction coefficient, the S-BAC method can enhance the prediction of the friction coefficient from texture parameters, compared with the von Meier and indentor methods.

The weak relationship may be because the pavement texture capture equipment can only obtain macrotexture. This study took measures expected to separate the contribution of macrotexture to friction during the friction measurement. However, microtexture dominates the low friction coefficient, which makes the friction coefficient unsensitive to the macrotexture difference.

## 5. Discussion

The von Meier, indentor, and S-BAC methods in this study emphasize different factors affecting tire-pavement contact. This may make the difference to the rubber penetration depth (Figures 15 and 17) and its effect on the relationship between the pavement texture and friction coefficient (Figure 20). The results obtained by the S-BAC and von Meier method are only close to each other under specific conditions. This study believes that the inherent parameter of the von Meier method causes the differences in penetration depth as the pavement morphologies in this study cover most of the pavement texture on asphalt concrete pavement (MTD ranges from 0.31~1.41 mm). Similarly, '$S$' in the indentor method was derived from the contact between the tire and the surfaces with wooden triangular elements (1 cm, 0.5 cm, and 2 cm separated from each other) [3], which causes the differences in rubber penetration depth [42].

For future studies, the S-BAC method proposed in this study could be improved in the following aspects:

Firstly, a thick rubber block should be chosen to simulate the contact between the tire tread and the pavement. The 6-mm rubber block may resemble non-mobile friction testers, such as the British pendulum tester and dynamic friction tester. However, tires would have a rubber tread of significantly more than 8 mm to accommodate a tread depth of 8 mm, and the tread sits on a rather flexible belt, so the 6-mm thick rubber block on a steel plate would not resemble an actual tire.

Secondly, pavement specimens bigger than 60 × 60 mm$^2$ should be used. In practice, the rubber blocks used in this study were plain, while the vehicle tires in the previous two methods had various patterns. These patterns divide the tire tread into smaller rubber blocks (smaller than the rubber block used in the S-BAC method). The von Meier and indentor methods used pavement profiles of 100 and 90 mm, respectively. Hence, a 60 × 60 mm$^2$ specimen may be sufficient for enveloping for car tires. However, the size is too small to represent macrotexture.

The specimen size should be about $0.3 \times 0.3$ m$^2$ to represent macrotexture according to the smallest acceptable size of specimens in ISO 13473-1.

In addition, the S-BAC method can also be extended to field tests with complex conditions if there are reliable methods to extract the real tire-pavement contact area from the nominal contact area. For example, Du et al. [14] and Woodward et al. [43] obtained the tire-pavement contact area by marking the abrasion area via colored powder or painting. After investigating the scale of the recognized contact area, the S-BAC method can be applied to obtain the penetration depth for dynamic contact, which can provide a helpful reference for road performance evaluation and prediction.

## 6. Conclusions

This study describes a new method to measure the rubber penetration depth on pavement based on the contact area measured using a staining method and the bearing ratio curve of the pavement surface. After discussing the factors that affect the test results of this method and comparing it with previous statistical methods, the following conclusions can be drawn:

- The contact area recognized by the staining method is at a scale of about 0.6 mm. Removing roughness with texture wavelengths shorter than 0.6 mm via filtering is better before measuring the bearing area curve. Otherwise, the penetration depth may be overestimated (but not by more than 5% for surface data with a sampling interval below 0.1 mm).
- The penetration depth obtained using the S-BAC method is close to that obtained using the von Meier method for some conditions. For surfaces with larger space between consecutive asperities, the penetration depth obtained using the S-BAC method is close to that using the von Meier method when the pressure increases, or the rubber hardness decreases.
- A strong linear correlation exists between the penetration depth obtained with the S-BAC method and the indentor method. The former is about 1.2~3 times higher than for the indentor method, and the multiple gradually increases with higher pressure or softer rubber.
- When calculating the root-mean-square height of the surface ($Sq$) and the root-mean-square slope of the surface ($Sdq$) based on texture data above the penetration depth, the relationship between these two pavement texture parameters and the low-speed friction coefficient is clearly enhanced after considering the penetration depth obtained via the S-BAC method.

Compared with the empirical methods, the S-BAC method did not depend on any controlling parameters from experience. It considers factors such as pressure and the properties of the rubber, which have a significant effect on the penetration depth. The experiment indicates that the S-BAC method strengthens the relationship between texture parameters and the low-speed friction coefficient. In future, the rubber and specimen size should be studied to represent the actual condition more reasonably, and exploring the application of the extended S-BAC method in field tests could be helpful. Furthermore, more reliable results may be obtained if the microtexture can be captured and considered in the evaluation process in future studies.

**Author Contributions:** Conceptualization, D.Y. and L.H.; methodology, D.Y. and C.T.; software, C.T.; validation, D.Y. and L.H.; data curation, J.G.; writing—original draft preparation, D.Y.; writing—review and editing, U.S.; visualization, D.Y. and M.R.; funding acquisition, X.Z. and L.H. All authors have read and agreed to the published version of the manuscript.

**Funding:** This research was funded by the Fundamental Research Funds for the Central Universities, CHD, grant number 300102213512; National Natural Science Foundation of China, China, grant number 52172392, 52268068; Key research and development Project of Hubei Province, China, grant number 2021BAA180.

**Institutional Review Board Statement:** Not applicable.

**Informed Consent Statement:** Not applicable.

**Data Availability Statement:** Data are contained within the article.

**Conflicts of Interest:** The authors declare no conflicts of interest.

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
