# Peer review of "A New Approach for Determining Rubber Enveloping on Pavement and Its Implications for Friction Estimation"

_coatings, doi:10.3390/coatings14030301_

Round 1

Reviewer 1 Report

Comments and Suggestions for Authors

The manuscript presents an in-depth study and a new methodology for estimating the depth of deformation of the tire upon contact with the surface texture of the road pavement.

The aim is to estimate the friction coefficient starting from the study of the depth of deformation of the tire.

In the manuscript, both the study of the state of the art on the investigated topic and the methodological approach used in the experimentation are excellent.

Although not fully achieving the proposed objective, as the title would suggest, the authors draw interesting conclusions and indicate future developments of the research.

In light of the foregoing, I would propose that the authors review the title of the manuscript, trying to give greater coherence to the results obtained.

Author Response

Thanks for the reviewer’s time and valuable suggestions. We have revised the title to “A new approach for determining rubber enveloping on pavement and its implications for friction estimation”.

Reviewer 2 Report

Comments and Suggestions for Authors

Comments in the attachment.

Reviewer 3 Report

Comments and Suggestions for Authors

The measurement of tire penetration into pavement holds significance for contact performance evaluation. This research introduces the S-BAC approach, leveraging the ratio of actual to nominal tire-pavement contact area (S) and the Bearing Area Curve (BAC), for assessing rubber penetration depth (d) on pavements. Utilizing contact between a smooth rubber block and pavement samples simulates tire-pavement contact. Following an analysis of influencing factors, the efficacy of the new method is compared with established techniques using d values and its application in correlating pavement texture parameters with friction. Findings indicate a linear relationship between d values obtained via S-BAC and conventional methods, yet discrepancies exist. The S-BAC method enhances the correlation between texture parameters and friction compared to alternatives, however, it has some flaws commented below:

How it is obtained rubber plate and what are the characteristics? Is it obtained commercially or experimentally? Please explain.

The new method is developed and the Manuscript is based on this but the software for image processing and filtering is not mentioned in the text. Which software is used for developing algorithms (named Method 1 and Method 2) presented in Figure 6? Algorithms must be explained.

Figure 18 is not mentioned in the Manuscript, and probably does not belong in this article. Please correct.
